# SMI-TED: A LARGE-SCALE FOUNDATION MODEL FOR MATERIALS AND CHEMISTRY

## ABSTRACT

We present SMI-TED (SMILE Transformer Encoder Decoder), a large-scale foundation model for materials and chemistry, trained on a massive dataset of 91 million SMILES samples (4 billion molecular tokens) from PubChem using self-supervised learning. Our encoder-decoder architecture enables a wide range of complex tasks, including the prediction of quantum chemical properties and reaction yields. We offer two model variants, with 289M and $8 \times 289M$ parameters, respectively, to accommodate different use cases. Our model achieves state-of-the-art results across multiple benchmark datasets, demonstrating its versatility and effectiveness. Notably, our model's latent space exhibits compositionality and separability, essential properties for higher-level reasoning tasks and few-shot learning capabilities. To facilitate further research and applications, we make our model weights and source code publicly available on HuggingFace and GitHub, respectively.

## 1 INTRODUCTION

Understanding molecular properties is crucial for accelerating discoveries in different fields, including drug development and materials science Pan (2023). Traditional methods rely on labor-intensive trial-and-error experiments, which are both costly and time-consuming Jablonka et al. (2024). However, recent advances in deep learning have enabled the use of foundation models to predict molecular properties and generate molecule candidates Flam-Shepherd et al. (2022); Wang et al. (2023); Wen et al. (2023), marking significant progress in scientific exploration.

The introduction of large-scale pre-training methodologies for chemical language models (LMs) represents a significant advancement in cheminformatics Sadybekov & Katritch (2023). These methodologies have demonstrated impressive results in challenging molecular tasks such as predicting properties and generating molecules Ross et al. (2022). The success of these models can be attributed to their ability to learn contextualized representations of input tokens through self-supervised learning on large unlabeled corpora Bommasani et al. (2021). This methodological approach typically involves two phases: pre-training on unlabeled data followed by fine-tuning on specific downstream task Yang et al. (2023). By reducing the reliance on annotated datasets, this approach has broadened our understanding of chemical language representations Guo et al. (2023).

Simplified Molecular-Input Line Entry System, SMILES, provide natural graphs that encode the connectivity information from the line annotations of molecular structures Li et al. (2022). SMILES defines a character string representation of a molecule by performing a depth-first pre-order spanning tree traversal of the molecular graph, generating symbols for each atom, bond, tree-traversal decision, and broken cycles Wei et al. (2023). Therefore, the resulting character string corresponds to a flattening of a spanning tree of the molecular graph. SMILES is widely adopted for molecular property prediction as SMILES is generally more compact than other methods of representing structure, including graphs Öztürk et al. (2020). There are billions of SMILES available on different open-sources repositories Tingle et al. (2023). However, most SMILES sequences do not belong to well-defined molecules Wigh et al. (2022). Alternative string-based representations exist, such as SELFIES. However, focusing on molecular optimization tasks on the learned representation space, suggested no obvious shortcoming of SMILES with respect to SELFIES in terms of optimization ability and sample efficiency Gao et al. (2022). The quality of the pre-training data plays a more important role on the outcome of the foundation model Wang et al. (2023); Takeda et al. (2023).

Towards this direction, we present a novel family of molecular encoder-decoder foundation models, denoted as SMI-TED$_{289M}$. Our SMI-TED$_{289M}$ encoder-decoder foundation model was obtained using a transformer-based molecular tokens encoder model aligned with an encoder-decoder mechanism trained on a large corpus of 91 million carefully curated molecules from PubChem Kim et al. (2023), resulting in 4 billion molecular tokens. Our main contributions are:

- We pre-train a large-scale family of encoder-decoder molecular open-source foundation models, denoted as SMI-TED$_{289M}$, on over 91 million molecules carefully curated from PubChem Kim et al. (2023), which is equivalent to 4 billion of molecular tokens.

- Our SMI-TED$_{289M}$ family of foundation models encompasses two distinct configurations: base, which has 289 million parameters; and the Mixture-of-O$_{SMI}$-Experts, MoE-O$_{SMI}$, characterized by a composition of $8 \times 289M$ parameters. Checkpoints for these models are fully accessible on HuggingFace: **suppressed for blind review**. Moreover, the source code is available at: **suppressed for blind review**.

- We perform extensive experimentation on several classification and regression tasks from 11 benchmark datasets, covering quantum mechanical, physical, biophysical, and physiological property prediction of small molecules. We also evaluate the reconstruction capacity of our SMI-TED$_{289M}$ considering the MOSES benchmarking dataset Polykovskiy et al. (2020). We also conducted high-throughput experiments on Pd-catalyzed Buchwald–Hartwig C–N cross-coupling reactions, predicting reaction yields. Furthermore, a study investigating the embedding created by SMI-TED$_{289M}$ and few-shot learning is also provided, indicating compositionality of the learned molecular representations.

Our results section demonstrates state-of-the-art performance of SMI-TED$_{289M}$ on different tasks, molecular properties prediction, molecule reconstruction, and an efficient metric for molecular latent space. Compositionality of the latent space suggests strong potential for chemical reasoning tasks. The SMI-TED$_{289M}$ family consists of two main variants (289M, and $8 \times 289M$), offering flexibility and scalability for different scientific applications.

## 2 OVERVIEW OF THE PROPOSED APPROACH

This section presents an overview of the proposed SMI-TED$_{289M}$ foundation model for small molecules. Here, we outline the process of collecting, curating, and pre-processing the pre-train data. Additionally, we describe the token encoder process and the SMILES encoder-decoder process. Finally, we explain the Mixture-of-O$_{SMI}$-Experts approach used to scale the base model. Fig. 1 illustrates the general architecture of the base model.

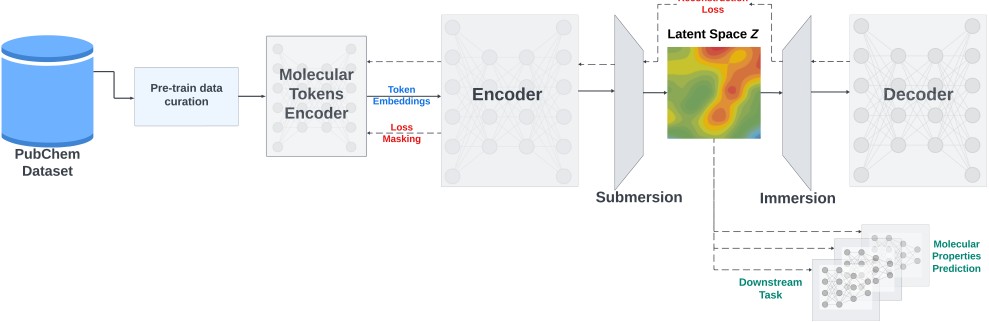

Figure 1: This figure illustrates the general architecture of the base SMI-TED$_{289M}$ model.

### 2.1 PRE-TRAINING DATA

The pretraining data originated from the PubChem data repository, a public database containing information on chemical substances and their biological activities Kim et al. (2023). Initially, 113 million SMILES strings were collected from PubChem. These molecular strings underwent deduplication and canonicalization processes to ensure uniqueness Heid et al. (2021). Subsequently,

a molecular transformation was conducted to verify the validity of the molecules derived from the unique SMILES strings, resulting in a set of 91 million unique and valid molecules.

To construct the vocabulary, we employed the molecular tokenizer proposed by Schwaller et al. (2019). All 91 million molecules curated from PubChem were utilized in the tokenization process, resulting in a set of 4 billion molecular tokens. The unique tokens extracted from the resulting output provided a vocabulary of 2988 tokens plus 5 special tokens. In comparison, MoLFormer, trained on 1 billion samples with minimal curation, presented a vocabulary of 2362 tokens using the same tokenization process Ross et al. (2022). This suggests an improvement in the vocabulary model due to our curation process.

## 2.2 MODEL ARCHITECTURE

We conduct training for SMI-TED$_{289M}$ model employing a deep-bidirectional-transformers-based encoder Devlin et al. (2019) for tokens and an encoder-decoder architecture to compose SMILES. The hyper-parameters of SMI-TED$_{289M}$ base model are detailed in Table 1

Table 1: SMI-TED$_{289M}$ base architecture specificity.

| Hidden size | Attention heads | Layers | Dropout | Normalization |
|---|---|---|---|---|
| 768 | 12 | 12 | 0.2 | LayerNorm |

| Vocab size | # SMILES | # Mol tokens | # Encoder | # Decoder | Total params |
|---|---|---|---|---|---|
| 2993 | 91M | 4T | 47M | 242M | 289M |

To optimize the relative encoding through position-dependent rotations $R_m$ of the query and keys at position $m$, the SMI-TED$_{289M}$ uses a modified version of the RoFormer Su et al. (2021) attention mechanism. These rotations can be implemented as pointwise multiplications and do not significantly increase computational complexity as shown in Eq. (1).

$$Attention_m(Q, K, V) = \frac{\sum_{n=1}^{N} \langle \varphi(R_m q_m), \varphi(R_n k_n) \rangle v_n}{\sum_{n=1}^{N} \langle \varphi(R_m q_m), \varphi(R_n k_n) \rangle} \qquad (1)$$

where $Q$,$K$,$V$ are the query, key, and value respectively, and $\varphi$ is a random feature map.

We start with a sequence of tokens extracted from SMILES, each embedded in a 768-dimensional space. The encoder-decoder layer is designed to process molecular token embeddings, represented as $\mathbf{x} \in \mathbb{R}^{D \times L}$, where $D$ denotes the maximum number of tokens and $L$ represents the embedding space dimension. We limited $D$ at 202 tokens, as 99.4% of molecules in the PubChem dataset contain fewer tokens than this threshold.

In encoder-only models, a mean pooling layer is typically employed to represent tokens as SMILES in the latent space. However, this approach is limited by the lack of a natural inversion process for the mean pooling operation. To overcome this limitation, we aim to construct a latent space representation for SMILES by submersing the $\mathbf{x}$ in a latent space, denoted as $\mathbf{z}$, as described in Eq. 2.

$$\mathbf{z} = (\text{LayerNorm}\,(\text{GELU}\,(\mathbf{x}\mathbf{W}_1 + \mathbf{b}_1)))\,\mathbf{W}_2, \qquad (2)$$

where $\mathbf{z} \in \mathbb{R}^L$, $\mathbf{W}_1 \in \mathbb{R}^{D \times L}$, $\mathbf{b}_1 \in \mathbb{R}^L$, $\mathbf{W}_2 \in \mathbb{R}^{L \times L}$, with $L$ denoting the latent space size (specifically, $L = 768$) and $D$ representing the original feature space size (namely, $D = 202$). Subsequently, we can immerse $\mathbf{z}$ back by calculating Eq. 3.

$$\hat{\mathbf{x}} = (\text{LayerNorm}\,(\text{GELU}\,(\mathbf{z}\mathbf{W}_3 + \mathbf{b}_3)))\,\mathbf{W}_4 \qquad (3)$$

where $\hat{\mathbf{x}} \in \mathbb{R}^{D \times L}$, $\mathbf{W}_3 \in \mathbb{R}^{L \times L}$, $\mathbf{b}_3 \in \mathbb{R}^L$, $\mathbf{W}_4 \in \mathbb{R}^{L \times D}$.

A language layer (decoder) is used to process $\hat{\mathbf{x}}$, where it applies non-linearity and normalization, and projects the resulting vector into a set of logits over the vocabulary, which can then be used to predict the next token in the molecular Ferrando et al. (2023).

## 2.3 PRE-TRAINING STRATEGIES

Pre-training of SMI-TED$_{289M}$ was performed for 40 epochs through the entire curated PubChem dataset with a fixed learning rate of 1.6e-4 and a batch size of 288 molecules on a total of 24 NVIDIA V100 (16G) GPUs parallelized into 4 nodes using DDP and *torch run*. It involves two distinct phases: i) Learning of token embeddings through a masking process; ii) Subsequently, the token embeddings are mapped into a common latent space that encapsulates the entire SMILES string. This latent space not only facilitates the representation of the SMILES but also enables the reconstruction of both individual tokens and complete SMILES strings. Consequently, the pre-training process involves two separate loss functions: one for the token embeddings, which is based on the masking process, and another for the encoder-decoder layer, which focuses on the reconstruction of tokens. Two pre-training strategies are employed:

- In phase 1, the token encoder is initially pre-trained using 95% of the available samples, while the remaining 5% is reserved for training the encoder-decoder layer. This partitioning is necessary as the token embeddings may encounter convergence difficulties in the initial epochs, which could adversely affect the training of the encoder-decoder layer.

- In phase 2, once the token embeddings layer has achieved convergence, the pre-training process is expanded to utilize 100% of the available samples for both phases. This approach leads to an enhancement in the performance of the encoder-decoder layer, particularly in terms of token reconstruction.

For encoder pre-training we use the masked language model method defined in Devlin et al. (2019). Initially 15% of the tokens are selected for possible learning. From that selection, 80% of the tokens are randomly selected and replaced with the [MASK] token, 10% of the tokens are randomly selected to be replaced with a random token, while the remaining 10% of the tokens will be unchanged.

The adoption of different pre-training strategies has proven instrumental in enhancing the efficiency of our model, as evidenced by improvements observed in the loss functions. For detailed insights into the loss functions and pre-training methodologies, refer to the Supplementary Materials.

## 2.4 MIXTURE-OF-O$_{\text{SMI}}$-EXPERTS

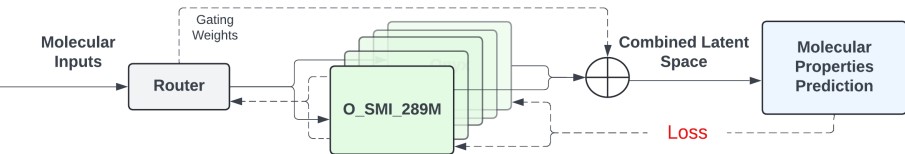

Figure 2: Mixture-of-O$_{\text{SMI}}$-Experts for downstream tasks.

The Mixture-of-O$_{\text{SMI}}$-Experts, MoE-O$_{\text{SMI}}$ comprises a set of $n$ "expert networks" labeled as $E_1, E_2, \ldots, E_n$, augmented through a gating network denoted as $G$, tasked with generating a sparse $n$-dimensional embedding space optimized for a downstream task as illustrated by Fig. 2.

Here, we map each SMILES into tokens and then convert the input tokens to the latent space. A mean pooling method is applied to all token embeddings in order to produce a meaningful embedding of the molecule. The architecture is equipped with a router module responsible for determining the $n$ experts that will be activated, refining the adaptability and specialization of the system. Let $G(x)$ and $E_i(\hat{x})$ denote the output of the gating network and the output of the $i$-th expert network, respectively, for a given input $\hat{x}$ of SMILES and $x$, which is the embeddings space, following a similar notation as proposed in Shazeer et al. (2017). The resulting output $y$ is defined as follows:

$$y = \sum_{i=1}^{n} G(x)_i E_i(\hat{x}) \tag{4}$$

The resulting embedding space $y$ is used to train a task-specific feed-forward network, where the loss function is chosen according to the studied downstream task. The optimization process refines the

parameters of $G(x)$. If the gating vector is sparse, we can use softmax over the Top-K logits of a linear layer Shazeer et al. (2017).

$$G(x) := Softmax(TopK(x \cdot Wg)) \tag{5}$$

where $(TopK(\ell))_i := \ell_i$ if $\ell_i$ is among the $TopK$ coordinates of logits $\ell \in \mathbb{R}^n$ and $(TopK(\ell))_i := \infty$ otherwise. The router layer retains only the top $k$ values, setting the remaining values to $-\infty$ (which effectively assigns corresponding gate values as 0). This sparsity-inducing step serves to optimize computational efficiency Jiang et al. (2024). Here, we define MoE-O$_{SMI}$ as $n = 8$ and $k = 2$, which means that MoE-O$_{SMI}$ is composed by $8\times$ SMI-TED$_{289M}$ models, which 2 models are activated through the router each round.

## 3 EXPERIMENTS

To evaluate the effectiveness of our proposed methodology, we conducted experiments using a set of 11 datasets sourced from MoleculeNet Wu et al. (2018) as demonstrated in Table 2. Specifically, we evaluated 6 datasets for classification task and 5 datasets for regression tasks. To ensure an unbiased assessment, we maintained consistency with the original benchmark by adopting identical train/validation/test splits for all tasks Wu et al. (2018). We also conducted the experiments considered 10 different seeds for all the tests in other to guarantee the robustness of the approach. Details are provided in the Supplementary Materials.

Table 2: Evaluated datasets description

| Dataset | Description | # compounds | # tasks | Metric |
|---|---|---|---|---|
| BBBP | Blood brain barrier penetration dataset | 2039 | 1 | ROC-AUC |
| HIV | Ability of small molecules to inhibit HIV replication | 41127 | 1 | ROC-AUC |
| BACE | Binding results for a set of inhibitors for $\beta$ – secretase 1 | 1513 | 1 | ROC-AUC |
| Clintox | Clinical trial toxicity of drugs | 1478 | 2 | ROC-AUC |
| SIDER | Drug side effect on different organ classes | 1427 | 27 | ROC-AUC |
| Tox21 | Toxicity measurements on 12 different targets | 7831 | 12 | ROC-AUC |
| QM9 | 12 quantum mechanical calculations | 133885 | 12 | Average MAE |
| QM8 | 12 excited state properties of small molecules | 21786 | 12 | Average MAE |
| ESOL | Water solubility dataset | 1128 | 1 | RMSE |
| FreeSolv | Hydration free energy of small molecules in water | 642 | 1 | RMSE |
| Lipophilicity | Octanol/water distribution coefficient of molecules | 4200 | 1 | RMSE |

To assess the reconstruction/decoder capacity of SMI-TED$_{289M}$ we considered the MOSES benchmarking dataset Polykovskiy et al. (2020). The MOSES dataset contains 1,936,962 molecular structures. For experiments, we consider the split proposed by Polykovskiy et al. (2020), where the dataset was divided into a training, test and scaffold test sets containing around 1.6M, 176k, and 176k molecules respectively. The scaffold test set contains unique Bemis-Murcko scaffolds that were not present in the training and test sets. We use this set to assess how well the model can generate previously unobserved scaffolds.

We also conducted high-throughput experiments on Pd-catalyzed Buchwald–Hartwig C–N cross-coupling reactions, measuring the yields for each reaction as described in Ahneman et al. (2018). The experiments utilized three 1536-well plates, covering a matrix of 15 aryl and heteroaryl halides, four Buchwald ligands, three bases, and 23 isoxazole additives, resulting in a total of 3,955 reactions. We employed the same data splits as in Ahneman et al. (2018) to assess our model's performance with training sets of varying sizes. An evaluation of the embedding space of SMI-TED$_{289M}$ is also provided, it uses the compositional molecules to evaluate the capability of the model to generate metric latent spaces.

## 4 RESULTS AND DISCUSSION

In this section, we present the analysis of results obtained using SMI-TED$_{289M}$ for different experiments conducted with various versions of the base model. We include: i) A study comparing frozen and fine-tuned versions of SMI-TED$_{289M}$; and a comparison with the State-of-the-Art (SOTA) on different benchmarking datasets for classification and regression molecular prediction tasks; ii)

An evaluation of MoE-O$_{SMI}$ for molecular properties prediction; iii) An evaluation of the Decoder module considering the MOSES benchmarking dataset; iv) A study comparing the latent space of SMI-TED$_{289M}$ based on compositional molecules metrics.

## 4.1 Comparison with SOTA on benchmarking tasks

**Results for classification tasks:** The analysis investigates the comparative efficacy of SMI-TED$_{289M}$ in its fine-tuned and frozen states versus state-of-the-art algorithms for molecular properties classification, as demonstrated in Table 3.

Table 3: Methods and Performance for the classification tasks of MoleculeNet benchmark datasets

| Method | Dataset | | | | | |
| --- | --- | --- | --- | --- | --- | --- |
| | BBBP | ClinTox | HIV | BACE | SIDER | Tox21 |
| GraphMVP Liu et al. (2021) | 72.4 ± 1.6 | 79.1 ± 2.8 | 77.0 ± 1.2 | 81.2 ± 0.9 | 63.9 ± 1.2 | 75.9 ± 0.5 |
| GEM Fang et al. (2022) | 72.4 ± 0.4 | 90.1 ± 1.3 | 80.6 ± 0.9 | 85.6 ± 1.1 | **67.2 ± 0.4** | 78.1 ± 0.1 |
| GROVER$_{Large}$ Rong et al. (2020) | 69.5 ± 0.1 | 76.2 ± 3.7 | 68.2 ± 1.1 | 81.0 ± 1.4 | 65.4 ± 0.1 | 73.5 ± 0.1 |
| ChemBerta Chithrananda et al. (2020) | 64.3 | 90.6 | 62.2 | - | - | - |
| ChemBerta2 Ahmad et al. (2022) | 71.94 | 90.7 | - | 85.1 | - | - |
| Galatica 30B Taylor et al. (2022) | 59.6 | 82.2 | 75.9 | 72.7 | 61.3 | 68.5 |
| Galatica 120B Taylor et al. (2022) | 66.1 | 82.6 | 74.5 | 61.7 | 63.2 | 68.9 |
| Uni-Mol Zhou et al. (2023) | 72.9 ± 0.6 | 91.9 ± 1.8 | **80.8 ± 0.3** | 85.7 ± 0.2 | 65.9 ± 1.3 | 79.6 ± 0.5 |
| MolFM Zhou et al. (2023) | 72.9 ± 0.1 | 79.7 ± 1.6 | 78.8 ± 1.1 | 83.9 ± 1.1 | 64.2 ± 0.9 | 77.2 ± 0.7 |
| MoLFormer Chang & Ye (2024) | 73.6 ± 0.8 | 80.5 ± 1.65 | 80.5 ± 1.4 | 86.3 ± 0.6 | 65.5 ± 0.2 | 80.46 ± 0.2 |
| SMI-TED$_{289M}$ (Frozen Weights) | 91.46 ± 0.47 | 93.49 ± 0.85 | 80.51 ± 1.34 | 85.58 ± 0.92 | 66.01 ± 0.88 | 81.53 ±0.45 |
| SMI-TED$_{289M}$ (Fine-tuned) | **92.26 ± 0.57** | **94.27 ± 1.83** | 76.85 ± 0.89 | **88.24 ± 0.50** | 65.68 ± 0.45 | **81.85 ± 1.42** |

Table 3 displays the performance of different advanced methods on different benchmarking datasets used for molecule classification tasks. SMI-TED$_{289M}$ consistently shows superior performance in four out of six datasets. Interestingly, using SMI-TED$_{289M}$ with its initial settings provided comparable results to SOTA methods available. However, fine-tuning SMI-TED$_{289M}$ further enhances its performance across all datasets. This indicates SMI-TED$_{289M}$' potential for accurate molecule classification, with potential for further optimization through fine-tuning. Detailed results for all the experiments are presented in the Supplementary Materials due to limit of pages.

**Results for regression tasks:** Next, we applied SMI-TED$_{289M}$ for prediction of chemical properties. The performance results across five challenging regression benchmarks, namely QM9, QM8, ESOL, FreeSolv, and Lipophilicity, are summarized in Table 4.

Table 4: Methods and Performance for the regression tasks of MoleculeNet benchmark datasets.

| Method | Dataset | | | | |
| --- | --- | --- | --- | --- | --- |
| | QM9 | QM8 | ESOL | FreeSolv | Lipophilicity |
| D-MPNN Yang et al. (2019) | 3.241 ± 0.119 | 0.0143 ± 0.0022 | 0.98 ± 0.26 | 2.18 ± 0.91 | 0.65 ± 0.05 |
| N-Gram Liu et al. (2019) | 2.51 ± 0.19 | 0.0320 ± 0.003 | 1.074 ± 0.107 | 2.688 ± 0.085 | 0.812 ± 0.028 |
| PretrainGNN Hu et al. (2019) | - | - | 1.100 ± 0.006 | 2.764 ± 0.002 | 0.739 ± 0.003 |
| GROVER$_{Large}$ Rong et al. (2020) | - | - | 0.895 ± 0.017 | 2.272 ± 0.051 | 0.823 ± 0.010 |
| ChemBERTa-2 Ahmad et al. (2022) | - | - | 0.89 | - | 0.80 |
| SPMM Chang & Ye (2024) | - | - | 0.818 ± 0.008 | 1.907 ± 0.058 | 0.692 ± 0.008 |
| MolCLR$_{GIN}$ Wang et al. (2022) | 2.357 ± 0.118 | 0.0174 ± 0.0013 | 1.11 ± 0.01 | 2.20 ± 0.20 | 0.65 ± 0.08 |
| Hu et al. Hu et al. (2020) | 4.349 ± 0.061 | 0.0191 ± 0.0003 | 1.22 ± 0.02 | 2.83 ± 0.12 | 0.74 ± 0.00 |
| MoLFormer Chang & Ye (2024) | 1.5894 ± 0.0567 | 0.0102 | 0.880 ± 0.028 | 2.342 ± 0.052 | 0.700 ± 0.012 |
| SMI-TED$_{289M}$ (Frozen Weights) | 7.4883 ± 0.0659 | 0.0179 ± 0.0004 | 0.7045 ± 0.0344 | 1.668 ± 0.0616 | 0.6499 ± 0.012 |
| SMI-TED$_{289M}$ (Fine-tuned) | **1.3246 ± 0.0157** | **0.0095 ± 0.0001** | **0.6112 ± 0.0096** | **1.2233 ± 0.0029** | **0.5522 ± 0.0194** |

Results presented in Table 4 indicates that SMI-TED$_{289M}$ presents superior results when compared to the state-of-the-art, outperforming its competitors in all the 5 datasets considered. To fine-tune SMI-TED$_{289M}$ is important to achieve state-of-the-art results in regression datasets, due to the complexity of such tasks. Table 4 elucidates the superiority of SMI-TED$_{289M}$ over the QM9 dataset. The QM9 dataset is composed by 12 tasks regarding to the quantum properties of molecules. A detailed overview over the results for QM9 are depicted in the next subsection. Detailed results for all experiments are in the Supplementary Materials of this paper.

**A deeper analysis over the QM9 benchmark:** In this subsection, we provide a deeper analysis over the results for the QM9 dataset. Table 5 details the results of the SOTA approaches each property

that composes QM9. Our comparative analysis extends to benchmarking the proposes encoder-decoder foundation model against state-of-the-art models derived from three distinct categories: (i) Graph-based, (ii) Geometry-based, and (iii) SMILES-based methodologies for prediction of molecular properties. The included baselines models are: 123-gnn Morris et al. (2019), a multitask neural net encoding the Coulomb Matrix (CM) Rupp et al. (2012), and its GNN variant as in the deep tensor neural net (DTNN) Schütt et al. (2017).

Table 5: Comparing state-of-the-art models performance over the QM9 dataset. **Blue** and **Orange** indicates best and second-best performing model, respectively.

| | Graph-based | | | Geometry-based | | | SMILES-based | |
|---|---|---|---|---|---|---|---|---|
| Measure | A-FP | 123-gnn | GC | CM | DTNN | MPNN | MoLFormer-XL | This paper |
| $\alpha$ | 0.49 | **0.27** | 1.37 | 0.85 | 0.95 | 0.89 | 0.33 | **0.27** |
| $C_v$ | 0.25 | **0.09** | 0.65 | 0.39 | 0.27 | 0.42 | 0.14 | **0.12** |
| $G$ | 0.89 | **0.05** | 3.41 | 2.27 | 2.43 | 2.02 | 0.34 | **0.11** |
| $gap$ | 0.0052 | 0.0048 | 0.01126 | 0.0086 | 0.0112 | 0.0066 | **0.0038** | **0.0036** |
| $H$ | 0.89 | **0.04** | 3.41 | 2.27 | 2.43 | 2.02 | 0.25 | **0.09** |
| $\epsilon_{homo}$ | 0.0036 | 0.0034 | 0.0072 | 0.0051 | 0.0038 | 0.0054 | **0.0029** | **0.0027** |
| $\epsilon_{lumo}$ | 0.0041 | 0.0035 | 0.0092 | 0.0064 | 0.0051 | 0.0062 | **0.0027** | **0.0026** |
| $\mu$ | 0.451 | 0.476 | 0.583 | 0.519 | **0.244** | **0.358** | 0.361 | 0.384 |
| $\langle R^2 \rangle$ | 26.84 | 22.90 | 35.97 | 46.00 | **17.00** | 28.5 | 17.06 | **14.72** |
| $U_0$ | 0.898 | **0.0427** | 3.41 | 2.27 | 2.43 | 2.05 | 0.3211 | **0.0850** |
| U | 0.89 | **0.111** | 3.41 | 2.27 | 2.43 | 2.00 | 0.25 | **0.0905** |
| ZPVE | 0.00207 | **0.0002** | 0.00299 | 0.00207 | 0.0017 | 0.00216 | 0.0003 | **0.0002** |
| Avg MAE | 2.6355 | 1.9995 | 4.3536 | 4.7384 | 2.3504 | 3.1898 | **1.5894** | **1.3246** |
| Avg std MAE | 0.0854 | 0.0658 | 0.1683 | 0.1281 | 0.1008 | 0.1108 | **0.0567** | **0.0157** |

Table 5 compares existing SOTA models in predicting quantum properties of molecules. The evaluation demonstrates that the proposed encoder-decoder foundation model outperforms current models in predicting 7 out of 12 quantum properties, and achieves either the best or second-best results in 11 out of 12 tasks.

However, when comparing with MoLFormer-XL, a model showing the second-best average error rate, it is noted that MoLFormer-XL's performance is influenced by its results on a specific property $\langle R^2 \rangle$. Although MoLFormer-XL performs well in average error rate, 123-gnn performs better in a larger number of tasks. In comparison, the proposed SMI-TED$_{289M}$ maintains consistent performance across all tasks, suggesting its robustness in predicting complex molecular properties.

## 4.2 MIXTURE-OF-O$_{SMI}$-EXPERTS PERFORM STUDIES

This study compare the results of MoE-O$_{SMI}$ against single SMI-TED$_{289M}$ models (frozen and fine-tuned). MoE-O$_{SMI}$ is composed by $8 \times 289M$ fine-tuned models for each specific task, we set $k = 2$, which means that 2 models are activated every step. The results for this study are shown in Table 6, which considers classification and regression tasks for molecular properties. Results refers to the best run of each version.

Table 6: MoE-O$_{SMI}$ and single SMI-TED$_{289M}$ models for molecular properties prediction.

| Method | Dataset | | | | | | | | |
|---|---|---|---|---|---|---|---|---|---|
| | BBBP↑ | ClinTox↑ | HIV↑ | BACE↑ | SIDER↑ | Tox21↑ | ESOL↓ | FreeSolv↓ | Lipo↓ |
| SMI-TED$_{289M}$ - Frozen | 92.27 | 95.02 | **81.81** | 87.18 | 67.11 | 82.22 | 0.6784 | 1.5832 | 0.6311 |
| SMI-TED$_{289M}$ - Fine-Tuned | 93.07 | **97.97** | 79.09 | 89.33 | 65.97 | 83.72 | 0.6024 | 1.2167 | 0.5413 |
| MoE-O$_{SMI}$ | **93.72** | 95.62 | 80.42 | **89.84** | **68.08** | **84.07** | **0.5566** | **1.1181** | **0.5376** |

Table 6 summarizes the performance metrics for each model across the different datasets. The results from the study indicate that MoE-O$_{SMI}$ consistently achieves higher performance metrics compared to single SMI-TED$_{289M}$ models (Frozen and Fine-Tuned) models across different tasks, especially in regression tasks where it improved results in all scenarios. These findings suggest that the MoE approach effectively leverages specialized sub-models to capture diverse patterns in the data, leading to improved accuracy in molecular property predictions. The mixture-of-experts approach serves as an efficient solution to scale single models and enhance performance for various tasks due to its ability to allocate specific tasks to different experts, optimizing single model's overall predictive capabilities.

### 4.3 REACTION-YIELD PREDICTION

Previously, we were able to show that the proposed SMI-TED$_{289M}$ model was able to perform compared to single tasks transformer-based methods. Chemical reactions in organic chemistry are described by writing the structural formula of reactants and products separated by an arrow, representing the chemical transformation by specifying how the atoms rearrange between one or several reactant molecules and one or several product molecules. Predicting outcomes of chemical reactions, such as their yield based on data gathered in high-throughput screening, is an important task in machine learning for chemistry.

We assessed this architecture against state-of-the-art methods using a high-throughput dataset of Buchwald–Hartwig cross-coupling reactions, focusing on predicting reaction yields Ahneman et al. (2018). This involves estimating the percentage of reactants converted into products. Our evaluation adhered to the schema and data divisions outlined in Ahneman et al. (2018). Table 7 presents the results for the SMI-TED$_{289M}$ model and compares its performance with existing state-of-the-art approaches.

| Subset/Split | DFT | Yield-BERT | Yield-BERT (Aug) | DRFP | YieldGNN | MSR2-RXN | SMI-TED$_{289M}$ |
|---|---|---|---|---|---|---|---|
| Rand 70/30 | 0.92 | 0.95±0.005 | 0.97±0.003 | 0.95±0.005 | 0.96±0.005 | 0.94±0.005 | **0.9841 ±0.0007** |
| Rand 50/50 | 0.9 | 0.92±0.01 | 0.95±0.01 | 0.93±0.01 | - | 0.93±0.01 | **0.982 ±0.0004** |
| Rand 30/70 | 0.85 | 0.88±0.01 | 0.92±0.01 | 0.89±0.01 | - | 0.90±0.01 | **0.979 ±0.0013** |
| Rand 20/80 | 0.81 | 0.86±0.01 | 0.89±0.01 | 0.87±0.01 | - | 0.87±0.01 | **0.976 ±0.0006** |
| Rand 10/90 | 0.77 | 0.79±0.02 | 0.81±0.02 | 0.81±0.01 | - | 0.80±0.02 | **0.961 ±0.0023** |
| Rand 5/95 | 0.68 | 0.61±0.04 | 0.74±0.03 | 0.73±0.02 | - | 0.69±0.03 | **0.912 ±0.0043** |
| Rand 2.5/97.5 | 0.59 | 0.45±0.05 | 0.61±0.04 | 0.62±0.04 | - | 0.57±0.05 | **0.875 ±0.0044** |
| Test 1 | 0.8 | 0.84±0.01 | 0.80±0.01 | 0.81±0.01 | - | 0.83±0.03 | **0.9832 ±0.0002** |
| Test 2 | 0.77 | 0.84±0.03 | 0.88±0.02 | 0.83±0.003 | - | 0.83±0.01 | **0.9820 ±0.0005** |
| Test 3 | 0.64 | 0.75±0.04 | 0.56±0.08 | 0.71±0.001 | - | 0.69±0.04 | **0.9827 ±0.0012** |
| Test 4 | 0.54 | 0.49±0.05 | 0.43±0.04 | 0.49±0.004 | - | 0.51±0.04 | **0.9825 ±0.0008** |
| Average 1-4 | 0.69 | 0.73 | 0.58±0.33 | 0.71±0.16 | - | 0.72±0.15 | **0.9826 ±0.0005** |

Table 7: Performance of SMI-TED$_{289M}$ compared with the state of the art in reaction-yield prediction on experimentally determined yields of Buchwald–Hartwig reactions through HTEs.

The results presented in Table 7 demonstrate the superiority of the proposed SMI-TED$_{289M}$ foundation model when benchmarked against state-of-the-art methods, including gradient-boosting and fingerprint-based approaches (DRFP) Probst et al. (2022), a DFT-based random forest model (DFT) Probst et al. (2022), and transformer-based models like Yield-BERT Schwaller et al. (2021) and its augmented variant, Yield-BERT(aug.) Schwaller et al. (2021), and MSR2-RXN Boulougouri et al. (2024). The performance of the Mamba-based model can be attributed to its pre-training on an expansive dataset of 91 million curated molecules, which provides a robust foundation of chemical knowledge that significantly enhances its predictive capabilities. This pre-training enables the model to achieve high accuracy even with limited training data, as evidenced by its sustained performance when trained on just 2.5% of the available samples—a scenario where task-specific models experience a marked decline in accuracy. To ensure the robustness of our model, we conducted each experiment with 10 different random seeds.

### 4.4 DECODER EVALUATION OVER MOSES BENCHMARKING DATASET

Next, we compared SMI-TED$_{289M}$ with different baseline models, such as the character-level recurrent neural network (CharRNN) Polykovskiy et al. (2020), SMILES variational autoencoder (VAE) Polykovskiy et al. (2020), junction tree VAE (JT-VAE) Jin et al. (2018), latent inceptionism on molecules (LIMO) Eckmann et al. (2022), MolGen-7b Fang et al. (2023), and GP-MoLFormer Ross et al. (2024). All baseline performances are reported on their corresponding test set consisting of 176k molecules. Standard metrics for evaluating model-generated molecules are reported in Table 8. All metrics are computed using MOSES.

When compared to baselines, SMI-TED$_{289M}$ is equally performant in generating unique, valid, and novel molecules that share high cosine similarity with the corresponding reference molecules at the fragment (Frag) level, consistent with low Fréchet ChemNet Distance (FCD). At the same time, SMI-TED$_{289M}$ generates molecules with high internal diversity (IntDiv), i.e., average pairwise dissimilarity. The scaffold cosine similarity (Scaf) and similarity to the nearest neighbor in the test set

Table 8: MOSES benchmarking dataset evaluation.

| Metric | Frag ↑ | Scaf ↑ | SNN ↑ | IntDiv ↑ | FCD ↓ |
|---|---|---|---|---|---|
| CharRNN | 0.9998 | 0.9242 | 0.6015 | 0.8562 | 0.0732 |
| VAE | 0.9984 | 0.9386 | 0.6257 | 0.8558 | 0.0990 |
| JT-VAE | 0.9965 | 0.8964 | 0.5477 | 0.8551 | 0.3954 |
| LIMO | 0.6989 | 0.0079 | 0.2464 | **0.9039** | 26.78 |
| MolGen-7b | 0.9999 | 0.6538 | 0.5138 | 0.8617 | **0.0435** |
| GP-MoLFormer | 0.9998 | 0.7383 | 0.5045 | 0.8655 | 0.0591 |
| SMI-TED$_{289M}$ | **0.9999** | **0.9999** | **0.9998** | 0.8565 | 1.1532 |

(SNN) of SMI-TED$_{289M}$ is superior to the baselines demonstrating that SMI-TED$_{289M}$ is effective in generating molecules of varying structures and quality compared to baseline methods.

### 4.5 LATENT SPACE STUDY

We conducted an experiment to investigate the structure of the latent space created by Large Language Models in the context of Chemistry. Molecular structures are composable from fragments, motifs, and functional groups. The composability of structure often translates into compositionality of structure-property relations, which is exemplified by powerful group contribution methods in chemical sciences. Compositionality of the learnt representation, however, does not follow automatically from the structure of the data and requires some combination of the learning architecture and learning constraints to emerge. Our approach was to utilize simple chemical structures that can be easily understood by humans, allowing us to anticipate relationships between elements, and examine the latent space for similar patterns. We constructed a dataset consisting of six families of carbon chains: $\mathcal{F} = \{CC, CO, CN, CS, CF, CP\}$. For each family, we generated a sequence of molecules by incrementally adding carbon atoms to the end of the SMILES string, up to a maximum of ten carbon atoms. For example, the family $CO$ consists of $\{CO, CCO, \cdots, CCCCCCCCCCO\}$. According to the domain expert's intuition consistent with the theory of chemical structure, in a metric space, such sequences should exhibit a hierarchical distance structure, where the distance between consecutive elements is smaller than the distance between elements with a larger difference in carbon count, i.e., $|\overline{C_n\mathcal{F}_i} - \overline{C_{n+1}\mathcal{F}_i}| < |\overline{C_n\mathcal{F}_i} - \overline{C_{n+2}\mathcal{F}_i}|$. Here, $n$ represents the number of carbon atoms, and $\overline{SMILE}$ denotes the projection of the SMILE string onto the embedding space.

First, we generated the embeddings for two different encoders, the MoLFormer and SMI-TED$_{289M}$, and used the t-SNEvan der Maaten & Hinton (2008) projection technique to generate pictures (Fig. 3) for visually inspecting the spaces. It is worth noting that the SMI-TED$_{289M}$ generated an embedding space that creates a nice separation of each family and respects the hierarchical distance structure, almost creating a linear relationship between each family. To quantify this relationship, we created a dataset of triples of SMILES, $\mathcal{T} = \{(C_n\mathcal{F}_{CC}, C_k\mathcal{F}_i, C_{n+k}\mathcal{F}_i) \mid 0 < n \leq 4, 0 < k \leq 5\}$, for the six families $\mathcal{F}_i$, resulting in six sub-datasets with 20 elements each, e.g., $(CC, CCO, CCCCO)$ is one element of the subset of type $CO$ where $n = 1, k = 2$. Then, we randomly selected one triple from each subset to feed a linear regression calculating $\alpha, \beta$, and $B_0$ such that $\alpha \cdot \overline{C_n\mathcal{F}_{CC}} + \beta \cdot \overline{C_k\mathcal{F}_i} + B_0 = \overline{C_{n+k}\mathcal{F}_i}$. We validated the linearity using the remaining 114 elements. The linear regression on the MoLFormer embeddings resulted in $R^2 = 0.55$ and $MSE = 0.237$, while on our model embeddings, it resulted in $R^2 = 0.99$ and $MSE = 0.002$.

We evaluated our encoder-decoder model using a few-shot learning process, where we input a few examples of triples, such as those mentioned earlier, to calculate $\alpha, \beta$, and $B_0$. We then use these parameters to generate embeddings for subsequent SMILES pairs and recreate the SMILES strings. To validate our approach, we tested the process on the same dataset of triples. We calculated the molecule similarity between the expected and generated results using the Tanimoto score (TS) Lipkus (1999). We repeated this test with different combinations of input triples, yielding similar results. For example, when using the input triples $[CC + CCCS = CCCCCS, \ CCCCC + CCCS = CCCCCCCCS]$ and querying all pairs in our subsets, we obtained a mean TS of 0.52. The top two similar results were $CC + CCCCCS = CCCCCS$ with TS = 0.92 and $CC + CCCCCO = CCCCCO$ with TS = 0.92, while the bottom two results were $CCCCC + CF = F[PH3+]F$ with TS = 0.06 and $CCCC + CF = F[PH3+]F$ with TS = 0.07.

Historically, group contribution was introduced in supervised learning context of structure-property relations. Our simple tests indicate that SMI-TED$_{289M}$ derived an equivalent of group contribution

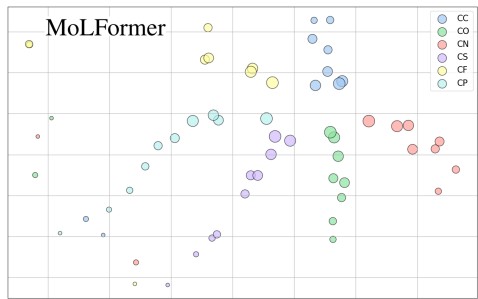 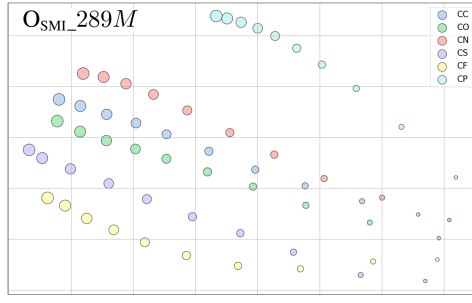

Figure 3: The figure shows the t-SNE projection of 60 small molecule embeddings. Color distinguishes between families, and point size represents the number of carbon atoms in the chain. Left: MoLFormer embeddings; Right: SMI-TED$_{289M}$ embeddings.

method purely from self-supervised learning of molecular structure. Signs of the emergence of compositionality of the learned molecular representations suggest strong potential of SMI-TED$_{289M}$ for reasoning applications. Further studies consistent with methodologies of compositionality analysis in natural languages are required to make stronger statements.

## 5 CONCLUSION

This paper introduces the SMI-TED$_{289M}$ family of chemical foundation models, which are pretrained on a curated dataset of 91 million SMILES samples from PubChem, amounting to 4 billion molecular tokens. The SMI-TED$_{289M}$ family includes two configurations: the base model with 289 million parameters and the MoE-O$_{SMI}$ model, which consists of $8 \times 289M$ parameters.

The performance of these models was evaluated through an extensive experimentation on different tasks, including molecular properties classification and prediction. Our approach achieved state-of-the-art results in most tasks, particularly in predicting molecular quantum mechanics, where it achieved the best or second-best results in 11 out of 12 tasks of the QM9 dataset.

One key observation is the model's robustness across various data splits for reaction-yield prediction, particularly in low-resource settings where only a small fraction of the dataset is used for training. This underscores the importance of leveraging large-scale pre-training to encode generalized chemical knowledge, which can then be fine-tuned for specific tasks like reaction yield prediction. In contrast, models that are tailored specifically for a given task tend to overfit to the nuances of the training data and struggle to generalize when the training set size is reduced, highlighting a critical limitation in their design.

We also investigated the structure of the latent space created by these language-based foundation models, using simple chemical structures for clarity. SMI-TED$_{289M}$ generated an embedding space that creates a nice separation of each family and respects the hierarchical distance structure, almost creating a linear relationship between each family. The encoder-decoder model's capabilities in few-shot learning were assessed by generating embeddings from a few example triples and using them to recreate SMILES strings, achieving a Tanimoto score of 0.92 in the best case.

The family of chemical foundation models presented in this paper offers flexibility and scalability for different scientific applications. Weights for the SMI-TED$_{289M}$ family of models are fully accessible on HuggingFace: **suppressed for blind review**. The source code is available at: **suppressed for blind review**.

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

## A   APPENDIX

## B   SUPPLEMENTARY MATERIALS

### B.1   DETAILED RESULTS - FROZEN WEIGHTS

Here, we provide the detailed results for every experiment conducted in this paper. First, we present the detailed results for the experiments considering frozen weights of SMI-TED$_{289M}$ for both, classification and regression tasks, considering the MoleculeNet benchmarking dataset. For SMI-TED$_{289M}$ frozen weights, we considered XGBoost Chen et al. (2015) as learner, and Optuna Akiba et al. (2019) for hyper-parameters optimization. Table 9 illustrates the results for the classification tasks using for 10 different seeds, and considering frozen weights.

Table 9: Classification results for 10 different seeds considering SMI-TED$_{289M}$ frozen weights.

| | ROC-AUC ↑ | | | | | |
|---|---|---|---|---|---|---|
| SEED | BBBP | HIV | BACE | SIDER | Clintox | Tox21 |
| 0 | 91.66 | 81.68 | 85.05 | 67.46 | 93.62 | 80.90 |
| 10 | 91.17 | 79.66 | 84.59 | 66.43 | 93.92 | 81.15 |
| 20 | 91.30 | 81.69 | 84.56 | 66.21 | 94.40 | 82.00 |
| 30 | 91.33 | 81.81 | 86.02 | 64.79 | 93.73 | 81.55 |
| 40 | 91.22 | 81.00 | 85.51 | 65.88 | 92.85 | 82.00 |
| 50 | 91.89 | 81.80 | 86.68 | 64.99 | 95.02 | 82.22 |
| 60 | 90.67 | 80.21 | 84.72 | 66.18 | 92.03 | 81.68 |
| 70 | 91.94 | 79.69 | 86.26 | 65.86 | 92.99 | 81.18 |
| 80 | 91.19 | 77.69 | 85.25 | 65.05 | 92.95 | 81.60 |
| 90 | 92.27 | 79.91 | 87.18 | 67.11 | 93.41 | 81.04 |
| Average | 91.46 | 80.51 | 85.58 | 66.00 | 93.49 | 81.53 |
| Std | 0.47 | 1.34 | 0.92 | 0.88 | 0.85 | 0.45 |

Table 10 elucidates the results for the regression tasks using for 10 different seeds, and considering frozen weights. Similar to the classification tasks, here we also use XGBoost as learner and Optuna for hyper-parameters optimization.

Table 10: Regression results for 10 different seeds considering SMI-TED$_{289M}$ frozen weights.

| | RMSE↓ | | | MAE↓ | |
|---|---|---|---|---|---|
| SEED | ESOL | FreeSolv | Lipophilicity | QM8 | QM9 |
| 0 | 0.6846 | 1.6248 | 0.6681 | 0.0184 | 7.4126 |
| 10 | 0.6784 | 1.7022 | 0.6400 | 0.0180 | 7.4956 |
| 20 | 0.6886 | 1.5832 | 0.6528 | 0.0174 | 7.6201 |
| 30 | 0.6880 | 1.7418 | 0.6311 | 0.0177 | 7.4845 |
| 40 | 0.7100 | 1.6443 | 0.6603 | 0.0185 | 7.5486 |
| 50 | 0.6933 | 1.6495 | 0.6515 | 0.0181 | 7.5118 |
| 60 | 0.6793 | 1.6285 | 0.6477 | 0.0182 | 7.5056 |
| 70 | 0.6884 | 1.7482 | 0.6411 | 0.0177 | 7.4128 |
| 80 | 0.7746 | 1.7468 | 0.6410 | 0.0179 | 7.4774 |
| 90 | 0.7599 | 1.6104 | 0.6654 | 0.0174 | 7.4135 |
| Average | 0.7045 | 1.6680 | 0.6499 | 0.0179 | 7.4883 |
| Std | 0.0344 | 0.0616 | 0.0120 | 0.0004 | 0.0659 |

## B.2 DETAILED RESULTS - FINE-TUNING

To fine-tune SMI-TED$_{289M}$, we used a fully connected network with 2 layers. Table 11 provides a detailed overview of the hyper-parameters considered for the fine-tuning of SMI-TED$_{289M}$. We used a single V100 NVIDIA (16G) GPU for the task. Detailed results considering SMI-TED$_{289M}$ for both, classification and regression tasks using the MoleculeNet benchmarking dataset are illustrated in Table 12 and Table 13. We run each task for 10 different seeds to guarantee the robustness of the results.

Table 11: SMI-TED$_{289M}$ fine-tuning architecture specificity.

| Hidden size | Attention heads | Layers | Dropout | Normalization |
|---|---|---|---|---|
| 768 | 12 | 12 | 0.2 | LayerNorm |

| Learning rate | # batch | # epochs | # tokens | # GPUs | Total params |
|---|---|---|---|---|---|
| 3e-5 | 32 | 500 | 202 | 1 NVIDIA V100 (32G) | 289M |

Table 12 presents the results BBBP, HIV, BACE, SIDER, Clintox, Tox21 datasets. For these classifications tasks, ROC-AUC has been defined as evaluation metric as in the MoleculeNet. We run each seed for 500 epochs.

Table 12: Classification results for 10 different seeds considering SMI-TED$_{289M}$ fine-tuning.

| SEED | ROC-AUC↑ | | | | | |
|---|---|---|---|---|---|---|
| | BBBP | HIV | BACE | SIDER | Clintox | Tox21 |
| 0 | 92.42 | 76.76 | 88.02 | 65.88 | 96.55 | 81.87 |
| 10 | 92.20 | 76.89 | 87.82 | 66.12 | 91.86 | 82.20 |
| 20 | 92.48 | 75.72 | 88.63 | 65.05 | 94.95 | 80.58 |
| 30 | 92.17 | 76.52 | 87.82 | 65.97 | 97.97 | 83.72 |
| 40 | 91.94 | 77.01 | 88.32 | 65.30 | 92.90 | 83.08 |
| 50 | 91.29 | 79.09 | 88.63 | 66.51 | 93.95 | 83.27 |
| 60 | 93.07 | 76.49 | 89.33 | 65.49 | 94.32 | 80.26 |
| 70 | 92.84 | 76.52 | 87.91 | 65.22 | 93.41 | 79.41 |
| 80 | 92.74 | 76.33 | 87.80 | 65.71 | 92.85 | 81.44 |
| 90 | 91.49 | 77.20 | 88.08 | 65.59 | 93.96 | 82.65 |
| Average | 92.26 | 76.85 | 88.24 | 65.68 | 94.27 | 81.85 |
| Std | 0.57 | 0.89 | 0.50 | 0.45 | 1.83 | 1.42 |

Results for ESOL, FreeSolv, Lipophilicity, QM8, and QM9 are presented in Table 13. As for classfication tasks, we also run each regression task for 10 different seeds, each one considering 500 epochs.

Table 13: Prediction results for 10 different seeds considering SMI-TED$_{289M}$ fine-tuning.

| SEED | RMSE↓ | | | MAE↓ | |
|---|---|---|---|---|---|
| | ESOL | FreeSolv | Lipophilicity | QM8 | QM9 |
| 0 | 0.6110 | 1.2258 | 0.5426 | 0.0092 | 1.2814 |
| 10 | 0.6110 | 1.2230 | 0.5375 | 0.0095 | 1.3371 |
| 20 | 0.6024 | 1.2230 | 0.5561 | 0.0094 | 1.3245 |
| 30 | 0.6124 | 1.2258 | 0.5472 | 0.0095 | 1.3291 |
| 40 | 0.6024 | 1.2258 | 0.5435 | 0.0095 | 1.3338 |
| 50 | 0.6024 | 1.2230 | 0.5413 | 0.0096 | 1.3302 |
| 60 | 0.6355 | 1.2167 | 0.5611 | 0.0099 | 1.3265 |
| 70 | 0.6116 | 1.2230 | 0.5513 | 0.0094 | 1.3293 |
| 80 | 0.6124 | 1.2258 | 0.5381 | 0.0095 | 1.3290 |
| 90 | 0.6110 | 1.2212 | 0.6029 | 0.0094 | 1.3249 |
| Average | 0.6112 | 1.2233 | 0.5522 | 0.0095 | 1.3246 |
| Std | 0.0096 | 0.0029 | 0.0194 | 0.0002 | 0.0157 |

QM9 and QM8 datasets contains 12 different metrics referring to the quantum properties of the molecules. Table 14 presents the results for the QM9 metrics: $\alpha$, $C_v$, $G$, $gap$, $H$, $\epsilon_{homo}$, $\epsilon_{lumo}$, $\mu$, $\langle R^2 \rangle$, $U_0$, $U$, $ZPVE$. Table 14 also show the avg MAE and avg std MAE. For each seed we considered 500 epochs.

Table 14: Prediction results over SMI-TED$_{289M}$ fine-tuning for QM9 dataset considering 10 different seeds.

| | QM9 | | | | | | | | | | | | |
|---|---|---|---|---|---|---|---|---|---|---|---|---|---|
| SEED | $\alpha$ | $C_v$ | $G$ | $gap$ | $H$ | $\epsilon_{homo}$ | $\epsilon_{lumo}$ | $\mu$ | $\langle R^2 \rangle$ | $U_0$ | $U$ | $ZPVE$ | Average |
| 0 | 0.2266 | 0.0893 | 0.1503 | 0.0035 | 0.0873 | 0.0025 | 0.0024 | 0.3859 | 14.2478 | 0.0919 | 0.0890 | 0.0002 | 1.2814 |
| 10 | 0.2898 | 0.1283 | 0.1276 | 0.0037 | 0.1126 | 0.0027 | 0.0025 | 0.3850 | 14.7824 | 0.1005 | 0.1093 | 0.0007 | 1.3371 |
| 20 | 0.2826 | 0.1226 | 0.0937 | 0.0036 | 0.0871 | 0.0026 | 0.0025 | 0.3846 | 14.7603 | 0.0737 | 0.0804 | 0.0005 | 1.3245 |
| 30 | 0.2827 | 0.1249 | 0.1270 | 0.0036 | 0.1088 | 0.0026 | 0.0026 | 0.3842 | 14.7041 | 0.1010 | 0.1069 | 0.0010 | 1.3291 |
| 40 | 0.2880 | 0.1351 | 0.1219 | 0.0043 | 0.1099 | 0.0035 | 0.0032 | 0.3853 | 14.7624 | 0.0935 | 0.0971 | 0.0019 | 1.3338 |
| 50 | 0.2832 | 0.1241 | 0.1042 | 0.0036 | 0.0816 | 0.0027 | 0.0025 | 0.3845 | 14.8141 | 0.0794 | 0.0814 | 0.0007 | 1.3302 |
| 60 | 0.2835 | 0.1263 | 0.0964 | 0.0036 | 0.0870 | 0.0027 | 0.0025 | 0.3850 | 14.7702 | 0.0785 | 0.0819 | 0.0007 | 1.3265 |
| 70 | 0.2873 | 0.1284 | 0.1014 | 0.0036 | 0.0864 | 0.0026 | 0.0027 | 0.3845 | 14.7972 | 0.0758 | 0.0810 | 0.0006 | 1.3293 |
| 80 | 0.2866 | 0.1270 | 0.0844 | 0.0036 | 0.0843 | 0.0027 | 0.0025 | 0.3842 | 14.8097 | 0.0752 | 0.0875 | 0.0007 | 1.3290 |
| 90 | 0.2829 | 0.1257 | 0.0957 | 0.0036 | 0.0874 | 0.0027 | 0.0025 | 0.3848 | 14.7414 | 0.0809 | 0.0907 | 0.0006 | 1.3249 |
| Average | 0.2793 | 0.1232 | 0.1103 | 0.0037 | 0.0932 | 0.0027 | 0.0026 | 0.3848 | 14.7190 | 0.0850 | 0.0905 | 0.0008 | 1.3246 |
| Std | 0.0187 | 0.0124 | 0.0205 | 0.0002 | 0.0120 | 0.0003 | 0.0002 | 0.0005 | 0.1688 | 0.0106 | 0.0107 | 0.0004 | 0.0157 |

Table 15 illustrates the results for the QM8 metrics: E1-CAM, E1-CC2, E1-PBE0, E2-CAM, E2-CC2, E2-PBE0, f1-CAM, f1-CC2, f1-PBE0, f2-CAM, f2-CC2, f2-PBE0. We also show the results for the average MAE and average std MAE. For both tasks, QM8 and QM9, our proposed SMI-TED$_{289M}$ demonstrated better results when compared to the state-of-the-art methods. To demonstrate the robustness and reliability of our approach we extensively evaluated it over 10 different seeds, considering 500 epochs for each seed.

Table 15: Prediction results over SMI-TED$_{289M}$ fine-tuning for QM8 dataset considering 10 different seeds.

| | QM8 | | | | | | | | | | | | |
|---|---|---|---|---|---|---|---|---|---|---|---|---|---|
| SEED | E1-CAM | E1-CC2 | E1-PBE0 | E2-CAM | E2-CC2 | E2-PBE0 | f1-CAM | f1-CC2 | f1-PBE0 | f2-CAM | f2-CC2 | f2-PBE0 | Average |
| 0 | 0.0040 | 0.0037 | 0.0037 | 0.0041 | 0.0050 | 0.0046 | 0.0081 | 0.0097 | 0.0078 | 0.0188 | 0.0226 | 0.0182 | 0.0092 |
| 10 | 0.0040 | 0.0039 | 0.0038 | 0.0043 | 0.0051 | 0.0053 | 0.0085 | 0.0100 | 0.0083 | 0.0195 | 0.0231 | 0.0186 | 0.0095 |
| 20 | 0.0040 | 0.0038 | 0.0037 | 0.0042 | 0.0050 | 0.0051 | 0.0084 | 0.0100 | 0.0082 | 0.0194 | 0.0231 | 0.0183 | 0.0094 |
| 30 | 0.0040 | 0.0038 | 0.0038 | 0.0043 | 0.0051 | 0.0053 | 0.0085 | 0.0100 | 0.0083 | 0.0195 | 0.0229 | 0.0185 | 0.0095 |
| 40 | 0.0041 | 0.0039 | 0.0039 | 0.0042 | 0.0051 | 0.0052 | 0.0084 | 0.0100 | 0.0081 | 0.0194 | 0.0230 | 0.0185 | 0.0095 |
| 50 | 0.0040 | 0.0039 | 0.0039 | 0.0043 | 0.0051 | 0.0053 | 0.0086 | 0.0100 | 0.0084 | 0.0195 | 0.0231 | 0.0185 | 0.0096 |
| 60 | 0.0043 | 0.0042 | 0.0042 | 0.0046 | 0.0054 | 0.0056 | 0.0091 | 0.0103 | 0.0085 | 0.0200 | 0.0235 | 0.0189 | 0.0099 |
| 70 | 0.0040 | 0.0038 | 0.0037 | 0.0042 | 0.0050 | 0.0050 | 0.0083 | 0.0101 | 0.0081 | 0.0193 | 0.0230 | 0.0186 | 0.0094 |
| 80 | 0.0040 | 0.0038 | 0.0038 | 0.0043 | 0.0051 | 0.0053 | 0.0084 | 0.0100 | 0.0083 | 0.0197 | 0.0230 | 0.0187 | 0.0095 |
| 90 | 0.0040 | 0.0038 | 0.0038 | 0.0042 | 0.0051 | 0.0051 | 0.0085 | 0.0101 | 0.0082 | 0.0194 | 0.0228 | 0.0183 | 0.0094 |
| Average | 0.0040 | 0.0039 | 0.0038 | 0.0043 | 0.0051 | 0.0052 | 0.0085 | 0.0100 | 0.0082 | 0.0194 | 0.0230 | 0.0185 | 0.0095 |
| Std | 0.0001 | 0.0001 | 0.0002 | 0.0001 | 0.0001 | 0.0003 | 0.0003 | 0.0001 | 0.0002 | 0.0003 | 0.0002 | 0.0002 | 0.0001 |

