# OpenReview forum: "SMI-TED: A large-scale foundation model for materials and chemistry"
_ICLR.cc/2025/Conference — ICLR 2025 Conference Withdrawn Submission_

### Official Review · Reviewer_1b8M · 2024-11-01

**Soundness:** 3
**Presentation:** 3
**Contribution:** 3
**Rating:** 5
**Confidence:** 2

**Summary:**

In this work, the authors introduce SMI-TED, a large-scale model trained on a model combined with a burst encoder and a decoder. The authors claim this model could be used as a foundation model in chemistry since it is trained on a large dataset, such as 91 million molecules taken from PubChem. Multiple benchmarks, including molecular property prediction, reaction yield prediction, and molecular reconstruction, are achieved by the model. The authors also state it is currently outperforming all other existing similar implementations. The model architecture combines a token encoder with an encoder-decoder mechanism, enabling both effective molecular representation learning and generation. A key finding is the model's ability to create well-structured latent spaces that capture meaningful chemical relationships and compositional properties, making it particularly effective for few-shot learning and chemical reasoning tasks. The paper is well-written and properly structured. In addition to the code, the model weights are also available as open source, which promotes open science.

**Strengths:**

The paper presents a large-scale model predominantly trained using PubChem molecules with a higher token space. The work uses an already established BERT model and combines a decoder model, thus forming an encoder-decoder architecture. If you look into the originality of this work mainly it focuses on large-scale training with a two-phase pre-training strategy on a massive curated dataset and its implementation of the MOE system for molecular modelling. The authors rigorously tested the model across 11 benchmark datasets and concluded that the model performs with high accuracy. While the clarity of the paper still could be improved, in some technical sections, the structure is presented well and the details and the code and model make this work much easier to reproduce compared to other works which sometimes lack the code. The authors deserve recognition for releasing their work as open-source, a choice that many still hesitate to make. This commitment not only fosters collaboration but also enhances accessibility for everyone. The model could be a valuable work in the field of chemistry.

**Weaknesses:**

While the paper has some merits several weaknesses should be addressed to improve the overall research work. The first thing is that the token space is quite confusing. Authors should clearly state how many unique tokens they got using the method they used for splitting the SMILES to get the tokens. It is somewhat confusing what the 4 billion molecular tokens are and whether it is even necessary to specify this multiple times. While the model shows impressive performance on standard benchmarks, it lacks critical error analysis and failure case studies that could provide a better understanding of the work. The work uses a preexisting set of models it is a bit unclear where the decoder comes from and the reason for choosing this architecture. Not clear what exactly is the innovation here. The ablation studies are insufficient to justify some of the architectural choices - particularly the selection of 289M parameters for the base model and the decision to use 8 experts in the MoE configuration. A clear justification for certain parameters are not given such as 40 epochs and 288 data points per batch. A learning rate of 1.6e-4 is used without explanation. The paper mentions using 24 NVIDIA V100 GPUs for training, but it lacks a detailed comparison of training time and resource requirements against baseline models like MoLFormer and Uni-Mol. PubChem has a large number of molecules but using that as a whole could cause data imbalance, this was not properly discussed and how the authors curated the dataset is not mentioned. No discussion of the potential over-representation of certain molecule types. Were any tests that were done using a diverse test dataset not clear?

**Questions:**

1. Authors should clearly state how many unique tokens they got using the method they used for splitting the SMILES to get the tokens. Why this method was used? Are there any other methods authors try?
2. Authors could elaborate on how they ended up with 4 billion tokens. If they simply state after splitting the 91 million SMILES strings they ended up with these many tokens it does not add any value to the paper. Mention how many unique tokens are used. What is the maximum length of the SMILES string used? Did they explore the unique token frequency?
3. How diverse is the chemical space?
4. Authors should explain their choice of model architecture. Did they consider training other models, such as the T5X model?
5. Not clear what exactly is the innovation in using this model and how one could practically benefit from this.
6. A more detailed justification of the parameters could be added to the paper.
7. While I appreciate the authors for making all the code and the model open source, they should understand that this is a double-blind review, and a simple Google search could easily reveal their work.
8. Authors could reduce using words like superior performance, state of the art, etc. Currently, this field is competitive there could be a better model in weeks.

---

### Official Review · Reviewer_DLgp · 2024-11-03

**Soundness:** 2
**Presentation:** 2
**Contribution:** 2
**Rating:** 3
**Confidence:** 4

**Summary:**

This paper introduces SMI-TED (SMILE Transformer Encoder Decoder), a foundation model for molecular property prediction and generation. At its core SMI-TED is an encoder-decoder architecture designed to create unified molecular representations - the encoder maps SMILES strings into a fixed-dimensional latent space that can be used for both downstream property prediction and molecule generation via the decoder. The model is pre-trained on 91 million curated SMILES strings from PubChem using a two-phase strategy: (1) masked language modeling for token embeddings, and (2) encoder-decoder training to learn compressed molecular representations that enable bidirectional mapping between SMILES strings and latent vectors. The architecture includes a base variant with 289M parameters and a Mixture-of-Experts variant with 8x289M parameters.

The authors evaluate their model on multiple benchmarks: (1) A comprehensive evaluation on MoleculeNet, including 6 classification and 5 regression tasks for molecular property prediction, (2) the MOSES benchmark for assessing molecular generation capabilities, and (3) a high-throughput reaction yield prediction dataset for evaluating reaction chemistry understanding.

The key contributions include: (1) Training and releasing large-scale foundation models trained on 91 million curated SMILES strings from PubChem, (2) developing a two-phase pre-training strategy combining masked language modeling and encoder-decoder training for learning molecular representations, (3) comprehensive evaluation across various molecular benchmarks and tasks, and (4) demonstrating the compositionality of the learned latent space using simple carbon chain molecules.

**Strengths:**

- The model is pre-trained on a large, curated dataset of 91 million SMILES strings from PubChem, amounting to 4 billion molecular tokens. The quality and scale of the pre-training data plays a vital role in the performance of foundation models, and the authors have done well to curate a high-quality dataset of substantial size.

- SMI-TED is thoroughly evaluated on a wide range of benchmark datasets covering various chemical tasks including molecular property prediction, molecule generation, and reaction yield prediction. Comprehensive experiments demonstrate the versatility and effectiveness of the model.

- The authors report standard deviations across multiple random seeds for their experiments, which is essential for assessing the robustness of the model's performance. This is a good practice that should be followed by all researchers to ensure the reproducibility of their results.

- The paper introduces two model variants - a base model with 289M parameters and a Mixture-of-Experts model with 8x289M parameters. This provides flexibility to accommodate different use cases and computational budgets. The MoE approach also represents an efficient way to scale the model to billions of parameters.

- The authors make their model weights and code publicly available, facilitating reproducibility and enabling other researchers to build upon their work. This commitment to open science is commendable.

- The analysis of the learned latent space and its compositional properties is interesting. The ability of SMI-TED's embeddings to capture meaningful chemical relationships, such as the hierarchical distance between molecules with increasing carbon chain lengths, suggests the model's potential for higher-level chemical reasoning tasks. This warrants further exploration.

**Weaknesses:**

**Novelty and Significance:**

- The proposed methodology of this paper is extremely limited in its novelty. Many of the components of the model, such as (1) a transformer trained on SMILES strings, (2) BERT-style pre-training, (3) utilization of sparsely-gated mixture-of-experts to scale the model, and (4) the general encoder-decoder architecture, have been explored in previous works. The only methodological novelties, to the best of my knowledge, are (1) the specific details behind the encoder-decoder training methodology, used to extract a fixed-dimensional latent space for the entire SMILES token sequence, and (2) the two-phase pre-training strategy combining masked language modeling and encoder-decoder training. These methodological novelties are quite minor and do not significantly advance the state-of-the-art in molecular property prediction or generation.

- For the two methodological novelties mentioned above, the authors provide limited description and discussion. Without more detailed explanations of these design choices, it is difficult to assess the significance of these contributions. As it stands, these design choices seem arbitrary. It seems that the authors intended to include more details for this in the supplementary material (see the quoted section below), but, as of the time of writing, the supplementary material does not include the promised information.

    > The adoption of different pre-training strategies has proven instrumental in enhancing the efficiency of our model, as evidenced by improvements observed in the loss functions. For detailed insights into the loss functions and pre-training methodologies, refer to the Supplementary Materials.

**Methodology:**

- The authors do not provide --- either in the main text or the supplementary material --- any information on how exactly the pre-trained model is fine-tuned. Is the output of the BERT-style encoder fed into prediction heads for each task? Or is the per-SMILES embedding generated by the encoder-decoder used for fine-tuning? This information is crucial for understanding the fine-tuning process and the model's performance on downstream tasks.

- The inclusion of the Mixture-of-Experts (MoE) model seems like an afterthought. The authors do not provide any results for the MoE model in the main results tables (3, 4, 5, 7, 8). The only mention of the MoE model is in table 6, where the authors report marginal improvements over the base model. Given that the MoE model has 8x the number of parameters, one would expect a more substantial improvement in performance. The authors should provide a more detailed analysis of the MoE model's performance and discuss the trade-offs between the base and MoE models.

**Experimental Evaluation:**

- For their QM9 evaluation, the authors claim (as shown in the quote below) that they compare their model against all relevant state-of-the-art models. However, the authors seemingly omit several state-of-the-art models that reportedly achieve better performance across some or all the tasks on the QM9 dataset. I have included a brief list of some of these models below.
    > Our comparative analysis extends to benchmarking the proposed encoder-decoder foundation model against state-of-the-art models derived from three distinct categories: (i) Graph-based, (ii) Geometry-based, and (iii) SMILES-based methodologies for prediction of molecular properties.

    **Pre-trained models missing from the comparison:**
    - Joint Multi-domain Pre-training (JMP) \[1\]
    - ET-OREO \[2\]
    - GNS+TAT+NN \[3\]

    **Scratch-trained models missing from the comparison:**
    - Equiformer \[4\]
    - MACE \[5\]
    - Allegro \[6\]

**Technical Clarity and Presentation:**

- Equations 2 and 3 are extremely confusing, and the dimensions do not match up.
    - Eq 2:
        $$\mathbf{z}=\left(\mathrm{LayerNorm}\left(\mathrm{GELU}\left(\mathbf{xW}_1+\mathbf{b}_1\right)\right)\right)\mathbf{W}_2$$

        If $\mathbf{x} \in \mathbb{R}^{D \times L}$ and $\mathbf{W_1} \in \mathbb{R}^{D \times L}$, then $\mathbf{x} \mathbf{W_1}$ doesn't make sense, as the dimensions don't match up. From my understanding, it seems that the operation that the authors are trying to perform is not exactly a simple matrix multiplication, but rather a tensor contraction described by the following einsum operation: `einsum("dl,dl->l", x, W_1)`. In mathematical notation, this could be written as $\text{diag}(\mathbf{x}^{\intercal} \mathbf{W}_1)$.

    - Eq 3:
        $$\hat{\mathbf{x}}=\left(\text{LayerNorm}\left(\text{GELU}\left(\mathbf{z}\mathbf{W}_3+\mathbf{b}_3\right)\right)\right)\mathbf{W}_4$$

        If $\mathbf{z} \in \mathbb{R}^{L}$, $\mathbf{W_3} \in \mathbb{R}^{L \times L}$, $\mathbf{b_3} \in \mathbb{R}^{L}$, and $\mathbf{W_4} \in \mathbb{R}^{L \times D}$, then $(\mathbf{z} \mathbf{W_3} + \mathbf{b_3}) \in \mathbb{R}^{L}$. Then, $(\mathbf{z} \mathbf{W_3} + \mathbf{b_3}) \mathbf{W_4} \in \mathbb{R}^{D}$, which doesn't match the reported dimensions of $\hat{\mathbf{x}} \in \mathbb{R}^{D \times L}$. Again, I believe the operation here would be $\mathbf{W}_4^{\intercal} \text{diag}(\mathbf{z} \mathbf{W}_3 + \mathbf{b}_3)$ (einsum: `einsum("l,ld->dl", z @ W_3 + b_3, W_4)`).

- Units are not provided for any of the reported metrics in the paper. This makes it difficult to interpret the results and compare them with other works. This is especially important for the QM9 benchmark.

- For the QM9 benchmark, the authors report an "average MAE" across all 12 tasks. However, each task has a different scale and range of values, so averaging the MAE across all tasks is not meaningful. If the authors want to report a single number for the QM9 benchmark, they should consider either (1) computing the rank of each model and reporting average ranks or (2) computing the average of the normalized MAE scores across all tasks.

**Nit Picks/Spelling/Grammar:**

- Bottom of page 3:
    > which can then be used to predict the next token in the **molecular**.

    Should be "molecule".


- Top of page 4:
    > Our comparative analysis extends to benchmarking the **proposes** encoder-decoder foundation model against state-of-the-art models derived from three distinct categories

    Should be "proposed".

---

\[1\] Shoghi, Nima, et al. "From molecules to materials: Pre-training large generalizable models for atomic property prediction." arXiv preprint arXiv:2310.16802 (2023).

\[2\] Feng, Rui, et al. "May the force be with you: Unified force-centric pre-training for 3d molecular conformations." Advances in Neural Information Processing Systems 36 (2024).

\[3\] Zaidi, Sheheryar, et al. "Pre-training via denoising for molecular property prediction." arXiv preprint arXiv:2206.00133 (2022).

\[4\] Liao, Yi-Lun, and Tess Smidt. "Equiformer: Equivariant graph attention transformer for 3d atomistic graphs." arXiv preprint arXiv:2206.11990 (2022).

\[5\] Kovács, Dávid Péter, et al. "Evaluation of the MACE force field architecture: From medicinal chemistry to materials science." The Journal of Chemical Physics 159.4 (2023).

\[6\] Musaelian, Albert, et al. "Learning local equivariant representations for large-scale atomistic dynamics." Nature Communications 14.1 (2023): 579.

**Questions:**

- Despite having "materials" in the title, the model is exclusively trained and evaluated on molecular data. The model cannot handle periodic boundary conditions or crystal structures, making it unsuitable for many materials science applications. If this is correct, then the title and framing of the paper is therefore misleading - this is a molecular foundation model, not a materials foundation model. Can the authors clarify the reasoning behind the title? If my understanding is not correct, can the authors provide some examples of materials science tasks that SMI-TED can handle? For example, how would SMI-TED handle bulk materials with periodic boundary conditions?

- Despite having 8x the number of parameters, the MoE model only shows seemingly marginal improvements over the base model (table 6). This does not seem to be a significant enough improvement to justify the additional computational cost. Can the authors shed some light on this? Also, why are all the main results tables (3, 4, 5, 7, 8) in the paper only showing results for the base model and not the MoE model?

- I am skeptical of the methodology used for converting between per-token embeddings and per-SMILES embeddings. I feel that this could be a potential bottleneck in the model's performance. Can the authors provide more details on this process and how it affects the model's performance? For example, I could imagine that using a much larger latent space dimension for the per-SMILES embeddings could improve the model's performance. Have the authors experimented with different latent space dimensions for the per-SMILES embeddings?

- Is there any reason why the decoder has nearly 5x the number of parameters as the encoder? For comparison, the BART-base encoder-decoder model \[1\] has ~82M encoder parameters and ~96M decoder parameters. The encoder-decoder ratio in SMI-TED is quite different. Can the authors provide some insights into this design choice?

- Does limiting $D$ (the maximum number of tokens) to 202 have any impact on the model's ability to encode longer SMILES strings (e.g., those for very large molecules or complex molecular structures)?


- The authors highlight the importance of having a natural inversion process as the motivation for their methodology from equations 2 and 3 (and thus the encoder-decoder architecture). This inversion is crucial for use cases such as molecule generation. However, for use-cases such as property prediction or latent-space exploration, the inversion process is less important. Have the authors experimented with using a simpler encoder-only architecture as a baseline for these use-cases (e.g., mean pooling, or using a `CLS`-token-like representation, a la BERT)? If so, what were the results?

- The authors claim that their vocabulary model is an improvement over MoLFormer because of their curation process. They use the fact that their model's vocabulary size is larger than MoLFormer's as evidence of this improvement. I'm not sure if I understand this reasoning. Could the authors provide more details on why a larger vocabulary size is indicative of a better vocabulary model?

    > The unique tokens extracted from the resulting output provided a vocabulary of 2988 tokens plus 5 special tokens. In comparison, MoLFormer, trained on 1 billion samples with minimal curation, presented a vocabulary of 2362 tokens using the same tokenization process Ross et al. (2022). This suggests an improvement in the vocabulary model due to our curation process.

---

\[1\] Lewis, M. "Bart: Denoising sequence-to-sequence pre-training for natural language generation, translation, and comprehension." arXiv preprint arXiv:1910.13461 (2019).

---

### Official Review · Reviewer_g5yf · 2024-11-03

**Soundness:** 2
**Presentation:** 3
**Contribution:** 2
**Rating:** 5
**Confidence:** 4

**Summary:**

The paper presents SMI-TED, a foundation model specifically designed for chemical and materials science applications. The model is pre-trained with an encoder-decoder architecture on a massive dataset of 91 million SMILES curated from Pubchem database. Two versions of the model is offered: a base model with 289 million parameters and a larger mixture-of-experts model(8x289M).

**Strengths:**

1. The structure of the paper is clear, enabling readers to readily identify the main take-away of the paper.

2. Curation of a Large Dataset: One of the paper's major contributions is the meticulous curation of a vast dataset from the Pubchem database for the training of foundation models. With 113 million SMILES strings reported from different sources, the effort required to curate, clean, and prepare such a dataset for effective training cannot be understated.

**Weaknesses:**

1. The paper's foundational motivation bears resemblance to the objectives of MolFormer, which similarly offers datasets for training a bio-chemical large language model. More thorough discussion is suggested to clarify the difference.

2. While the technical advancements presented are noteworthy, they appear incremental as they leverage pre-existing techniques such as pre-training and the Mixture-of-expert LLM. The authors could strengthen their contribution by elucidating the novel aspects of their approach.

3. Authors are encouraged to show more details about the training part such as loss curves, which provide a more granular view of the model's performance during training, thus offering valuable insights for fellow researchers.

**Questions:**

1. How concerned are the authors about the risk of data leakage? While the model may not encounter the exact same molecule-label pair during training, it could still encounter a very similar one. This possibility arises because there is no assurance that a specific molecule appears exclusively in the training or testing set.

2. What is the exact fine-tuning setup? From reading the manuscript, it is not clear to me if the fine-tuning is performed per task or once for all chemical datasets.

3. What is the exact process of training the MoE model? In Figure 2, the loss is designed for molecule property prediction, but in the manuscript, the author stated that "the loss function is chosen according to the studied downstream task." The authors are encouraged to clarify.

4. The results in Table 3 are intriguing. In particular, it is notable that the model’s performance on the HIV dataset decreases after fine-tuning. Could the authors provide insights into the factors contributing to this drop in performance?

5. In addition to the Buchwald-Hartwig (BH) dataset, other yield prediction datasets are available, such as the Suzuki-Coupling and USPTO datasets. The Suzuki-Coupling dataset offers another high-throughput option, and the USPTO dataset presents a unique challenge. It would be beneficial for the authors to consider evaluating their model on these datasets to further demonstrate its robustness and versatility.

[1]. Suzuki–Miyaura cross-coupling optimization enabled by automated feedback
[2]. Prediction of Chemical Reaction Yields using Deep Learning

---

### Official Review · Reviewer_r1RX · 2024-11-03

**Soundness:** 2
**Presentation:** 3
**Contribution:** 2
**Rating:** 3
**Confidence:** 4

**Summary:**

This paper proposes SMI-TED, a transformer-based foundation model for property predictions and reaction yields. The model is pretrained on a large dataset consisting of 91 million molecules in a self-supervised learning manner, using SMILES as representation. The authors propose two configurations: a base model with 289M parameters and a mixture-of-experts model. Experimental results on various molecular prediction tasks are conducted to demonstrate the effectiveness of the proposed model.

**Strengths:**

1. The paper curated a dataset of 91 million SMILES representations from PubChem. This large-scale dataset can generally help further develop large foundation models.

2. The authors have promised open access to the model and code.

3. The paper is well-written and easy to follow.

**Weaknesses:**

1. My major concern about the paper is the lack of novelty. The core of the paper is a standard transformer architecture with very limited modifications for molecular data. This choice of a transformer-based encoder-decoder setup for SMILES has been extensively applied to similar tasks. I did not notice clear statements or claims on substantial changes/improvements in the network architecture or the input representation.

2. Since the technical novelty is minor, I believe most performance improvement comes from using a large dataset. However, I did not spot any significant challenge in curating such a dataset since all data is publically available at PubChem.

3. The experimental results only show limited improvements over baseline methods on standard benchmarks. For MoleculeNet, the performance is very close to and sometimes even worse than state-of-the-art methods such as UniMol, whose results are missing for regression tasks. Additionally, since these methods are not trained on such a large dataset, I think they will surpass the proposed method if they were trained on the same data scale.

**Questions:**

1. Please change all citations from \cite to \citep

2. What is the purpose of the “molecule reconstruction” task? I cannot relate this task to any real-world applications.

3. In Figure 1, please replace “submersion” and “immersion” with simpler terms. The use of these advanced mathematical concepts is overly complex and imprecise for what could be expressed as straightforward transformation layers.

---

### Official Review · Reviewer_2n3q · 2024-11-03

**Soundness:** 2
**Presentation:** 2
**Contribution:** 2
**Rating:** 5
**Confidence:** 4

**Summary:**

* This paper introduces SMI-TED, an encoder-decoder Transformer-based model for learning representations of molecules that can be used for downstream tasks such as reaction yield prediction.
* SMI-TED is trained in a multi-step process (Section 2.3). First, the encoder is trained in a BERT-like self-supervised manner, before the decoder and encoder are trained with a reconstruction loss only (although I was not totally sure this was the only loss used in this part?). Finally, the weights can be fin- tuned for specific regression or classification tasks.
* The authors develop a method for combining the representations of multiple models into a single one using a learnt gating function (Section 2.4) and show how this ensembling leads to better downstream classification performance (Section 4.2).
The authors evaluate SMI-TED on several different regression/classification tasks, including MoleculeNet (Wu et al., 2018) tasks.
* Finally, the authors analyze how different molecules are organized in SMI-TED’s latent space (Section 4.5).

## Summary of my review.
While SMI-TED obtains excellent empirical performance across a range of different molecular tasks, I felt parts of the experimental setup were hard to follow and it was difficult to evaluate how the different design choices made contributed to model performance. I have therefore gone with a lower overall score, but am happy to reconsider if these points get addressed satisfactorily in the rebuttal.

**Strengths:**

# S1 Superior empirical performance on several different tasks
SMI-TED generally obtains better (and when not strictly "better", often at least comparable) performance to a wide range of different benchmarks across a diverse range of different tasks. While I have some questions about some of these results (see W2), this superior performance suggests the representations learnt by the pre-training regime proposed would be useful for others.

# S2 Authors commit to releasing their code and model weights
While currently omitted for blind review, the code and checkpoints that the authors have committed to release should allow reproducibility and adaptation of the model by other researchers.

# S3 Interesting experiment in Table 7 showing the benefits of pre-training in low data regimes
In Table 7 (evaluating performance on a reaction yield task), the authors demonstrate that the SMI-TED model does well even when the amount of training data is restricted. This is different to the baselines considered, where performance degrades significantly as the training data is reduced. This highlights the advantage of the representations learnt and suggests the model might be helpful in important and often encountered low data regimes.

**Weaknesses:**

# W1 Hard to know from experiments the importance of different design choices
While the authors include a useful ablation study on ensembling and fine-tuning (Section 4.2), the effect of the decoder and the second pre-training phase does not seem to be empirically evaluated as far as I can tell, despite its importance being emphasized (line 180).

# W2. Experimental setup is confusing in places
I found the experimental setup hard to follow in places. The metric considered is sometimes not defined (e.g., Table 7 — are these $R^2$ values or something else being shown?), and sometimes also the experimental procedure is unclear (e.g., I did not understand the procedure for generating “novel” molecules in Section 4.4).

Furthermore, I found the differences and discrepancies to values reported in earlier work confusing (or even between the similar experiments in this paper, e.g. Table 5 and 6). For instance, in Table 3 MolFormer is reported as obtaining a score of 73.6 on the BBBP task, although the original paper (which is actually citation Ross et al., 2022 rather than the Chang and Ye, 2024 used here?) quotes a score of 93.7 on this task (Table 1, Ross et al., 2022). Perhaps experimental differences can explain these discrepancies, but it would be useful to have these clarified.

# W3. Limited novelty
The approach seems similar to MolFormer (Ross et al., 2022): pre-training of a Transformer model using a BERT-like approach for downstream adaptation for various molecular regression/classification tasks.
The main differences seem to be the additional decoder (and reconstruction loss) and the ensembling of different representations? (Moreover, the ensembling does not actually seem to be used for many of the tasks?)


# W4. (minor — should be very easy to fix) Missing references/typos etc
Some examples:
* Line 51: Missing reference to SELFIES.
* Lines 187-189: This section discusses details of the pre-training material that are in the appendix. However, I was not able to find these. What part of the appendix is meant here?
* Line 223: Think $-\infty$ is meant here instead? Also what is $g$?
* Line 448: “The composabilitly of structure often translates into compositionality of structure-property relations”. It would be nice to have a reference or justification for this claim. (Many interesting properties are not simple functions of the different functional groups).
* Line 446: Carbon atoms are added to the start of the SMILES, not the end?

**Questions:**

Aside from those listed in the weaknesses section above, I had the following further questions:

Q1 How important is the reconstruction loss and second stage of pre-training for obtaining useful representations?

Q2 Lines 111-116 discuss needing more tokens than MolFormer, despite a smaller training set size; what do the extra tokens used here represent?

Q3 What does “convergence difficulties” mean on line 176?

**Details Of Ethics Concerns:**

Paper seems to highly overlap with another I am reviewing this year (so much so that portions of the text are identical between the papers, even if the models proposed are slightly different). (Have made a separate comment to the AC so that they can follow up on this).

---

### Note · Authors · 2024-11-25

I have read and agree with the venue's withdrawal policy on behalf of myself and my co-authors.